# 2D/2D Heterojunction of TiO_2_ Nanoparticles and Ultrathin G-C_3_N_4_ Nanosheets for Efficient Photocatalytic Hydrogen Evolution

**DOI:** 10.3390/nano12091557

**Published:** 2022-05-04

**Authors:** Ruifeng Du, Baoying Li, Xu Han, Ke Xiao, Xiang Wang, Chaoqi Zhang, Jordi Arbiol, Andreu Cabot

**Affiliations:** 1Catalonia Energy Research Institute—IREC, Sant Adrià de Besòs, 08930 Barcelona, Spain; ruifengdu@irec.cat (R.D.); kexiao@irec.cat (K.X.); xwang@irec.cat (X.W.); czhang@irec.cat (C.Z.); 2Departament d’Enginyeria Electrònica i Biomèdica, Universitat de Barcelona, 08028 Barcelona, Spain; 3Shandong Provincial Key Laboratory of Molecular Engineering, State Key Laboratory of Biobased Material and Green Papermaking, School of Chemistry and Chemical Engineering, Qilu University of Technology, Shandong Academy of Sciences, Jinan 250353, China; 4Catalan Institute of Nanoscience and Nanotechnology (ICN2), CSIC and BIST, Campus UAB, Bellaterra, 08193 Barcelona, Spain; xuhan@irec.cat (X.H.); arbiol@icrea.cat (J.A.); 5ICREA, Pg. Lluis Companys 23, 08010 Barcelona, Spain

**Keywords:** hydrogen evolution, 2D/2D heterojunction, charge separation

## Abstract

Photocatalytic hydrogen evolution is considered one of the promising routes to solve the energy and environmental crises. However, developing efficient and low-cost photocatalysts remains an unsolved challenge. In this work, ultrathin 2D g-C_3_N_4_ nanosheets are coupled with flat TiO_2_ nanoparticles as face-to-face 2D/2D heterojunction photocatalysts through a simple electrostatic self-assembly method. Compared with g-C_3_N_4_ and pure TiO_2_ nanosheets, 2D/2D TiO_2_/g-C_3_N_4_ heterojunctions exhibit effective charge separation and transport properties that translate into outstanding photocatalytic performances. With the optimized heterostructure composition, stable hydrogen evolution activities are threefold and fourfold higher than those of pure TiO_2,_ and g-C_3_N_4_ are consistently obtained. Benefiting from the favorable 2D/2D heterojunction structure, the TiO_2_/g-C_3_N_4_ photocatalyst yields H_2_ evolution rates up to 3875 μmol·g^−1^·h^−1^ with an AQE of 7.16% at 380 nm.

## 1. Introduction

Owing to the abundance of low-cost solar energy, the numerous uses of hydrogen and its advantages as an energy carrier, the photocatalytic generation of hydrogen is a highly appealing process [1,2]. However, the cost-effective photogeneration of hydrogen requires high activity and stable photocatalysts, development of which has been a long-standing goal. Over the past decades, numerous semiconductors have been tested as photocatalysts for hydrogen evolution. Among them, titanium dioxide (TiO_2_) has received special attention owing to its stability, high abundance, low toxicity, being the earliest to be discovered and becoming the first to be industrialized [3]. Nevertheless, due to its wide bandgap and relatively fast charge recombination rate, its applicability has been strongly limited. Numerous strategies have been proposed to improve the photocatalytic performance of TiO_2_, facilitating charge separation and promoting efficiency and activity, [4,5,6,7] including the control of its particle facets and morphology [8,9,10,11], its modification with cocatalysts [12,13,14,15] and its coupling with other semiconductors to form heterostructures [16,17,18,19,20,21,22,23,24].

Graphite carbonitride (g-C_3_N_4_) with a layered structure similar to graphite, high chemical stability and low cost has received increasing interest in recent years [25,26,27]. In particular, as a polymeric semiconductor, g-C_3_N_4_ has been recently reported as a promising candidate photocatalyst due to its unique structure and electronic characteristics, with a 2.7 eV bandgap that allows absorbing part of the visible spectrum [28,29]. Additionally, two-dimensional (2D) g-C_3_N_4_ nanosheets, benefiting from a huge specific surface area and a suitable band structure, have shown especially interesting properties and offer an excellent platform to produce heterojunctions with other semiconductors [30,31,32,33].

Recently, 2D/2D heterojunctions have been demonstrated to provide great advantages to improve charge separation [34,35]. 2D/2D heterojunctions simultaneously maximize the interface and surface areas, i.e., the charge transfer between the two materials and the interaction with the media, which can potentially improve photocatalytic activities.

In the present work, we target improving photocatalytic hydrogen production using 2D/2D heterojunctions. In this direction, we report the first synthesis of 2D/2D TiO_2_/g-C_3_N_4_ heterostructures. Such composite materials are produced from the electrostatic assembly of 2D anatase TiO_2_ flat nanoparticles synthesized through a simple colloidal method with 2D ultrathin g-C_3_N_4_. The produced heterostructures are tested as photocatalysts for hydrogen evolution under simulated solar light irradiation. The excellent hydrogen evolution performance obtained after optimizing the weight contents of TiO_2_ and g-C_3_N_4_ within 2D/2D heterojunction are rationalized using photoluminescence, photocurrent and impedance spectroscopy analysis.

## 2. Experiment

Synthesis of bulk g-C_3_N_4_ (bCN) and ultrathin g-C_3_N_4_ (uCN): Bulk g-C_3_N_4_ powder was synthesized by thermal polymerization of urea. Briefly, 10 g of urea (99%, Acros Organics) was placed into a ceramic crucible. The crucible was covered and heated to 550 °C at a ramp rate of 2 °C min^−1^ for 4 h under air atmosphere. After cooling to room temperature, the resulting light-yellow solid was ground with the mortar to obtain the bulk g-C_3_N_4_ powder. To obtain ultrathin g-C_3_N_4_ (uCN), bulk g-C_3_N_4_ (2.0 g) was placed in a covered ceramic crucible, and it was heated to 520 °C with a ramp rate of 5 °C min^−1^ for 2 h under air atmosphere to obtain a light-yellow powder.

Synthesis of TiO_2_ nanosheets: Titanium dioxide nanoparticles were prepared using a colloidal method. All the syntheses were performed using standard airless techniques [36,37]. Typically, 10 mL of oleylamine (OAm, 80–90%, Acros Organics, Geel, Belgium), 10 mL of octadecene (ODE, 90%, Sigma-Aldrich, Burlington, MA, USA) and 1 mL of oleic acid (OAc, 90%, Sigma-Aldrich, Burlington, MA, USA) were loaded in a three-neck flask and degassed under vacuum at 120 °C for 1 h while being strongly stirred using a magnetic bar. Then, 300 mg of TiF_4_ (99%, Sigma, Burlington, MA, USA) was added in a mixed solution of 2 mL OAm, 3 mL OAc and 6 mL ODE and sonicated for 0.5 h to prepare a precursor solution. Subsequently, under nitrogen atmosphere, 10 mL of the precursor solution were slowly added to the reaction flask, which was then heated to 290 °C at a rate of 5 °C min^−1^ and maintained for 1 h. The solid product was centrifuged and washed with acetone and hexane three times. The particles were finally dispersed in hexane at a concentration of 10 mg/mL.

Ligand removal from TiO_2_ nanoparticles: In a typical process, 10 mL of a TiO_2_ dispersion in hexane (2 mg/mL) was combined with 10 mL acetonitrile to form a two-phase mixture. Then, 1 mL of a HBF_4_ solution (48%, Sigma-Aldrich, Burlington, MA, USA) was added. The resulting solution was sonicated until the particles transferred from the upper to the bottom layer. The surface-modified particles were washed with ethanol and a 1 mol/L sodium hydroxide (85%, Sigma-Aldrich, Burlington, MA, USA) aqueous solution three times to remove the residual fluoride ions and ligands. The particles were then washed with water to adjust the PH close to neutral. Finally, the particles were dispersed in 10 mL of water with a small amount of DMF.

Synthesis of 2D/2D TiO_2_/ultrathin g-C_3_N_4_ (TiO_2_/uCN) composite: TiO_2_/uCN heterojunctions were produced by an electrostatic self-assembly method. Briefly, 20 mg of as prepared ultrathin g-C_3_N_4_ was dissolved in 10 mL of ultrapure water and sonicated for 1 h. The solution was then mixed with an ethanol solution of ligand-removed TiO_2_ nanoparticles with a weight ratio of 1:2, 1:1 and 2:1. The mixed solution was stirred for 24 h after 1 h of sonication. The obtained composite was collected by centrifuging, it was washed with ethanol 2 times, and it was finally dried at 60 °C for 12 h. The collected materials were named T_1_/uCN_2_, T_1_/uCN_1_ and T_2_/uCN_1_ based on the different TiO_2_/ultrathin g-C_3_N_4_ weight ratios. TiO_2_/bulk g-C_3_N_4_ (T/bCN) samples were prepared using the same procedure. For photocatalytic measurements, 1 wt% of Pt was loaded on the surface of the photocatalysts by a photoreduction method.

### Photocatalytic Hydrogen Evolution Procedure

The photocatalytic hydrogen evolution experiments were carried out in a Perfect Light Labsolar-III (AG) photoreactor (Pyrex glass) connected to a closed-loop gas circulation system. In a typical experiment, 20 mg photocatalyst was dispersed in 100 mL aqueous solution containing 10 mL methanol and 1 wt% Pt cocatalyst (40 uL 25.625 mmol/L H_2_PtCl_6_ aqueous solution). The mixed solution was bubbled with N_2_ for 30 min to ensure anaerobic state and illuminated 30 min with UV light before simulated solar light irradiation to ensure the complete loading of Pt. The incident light was provided by a 300 W Xe lamp with an AM 1.5 filter, and the reaction conditions were kept at room temperature. The resulting gas was analyzed by a Labsolar-III (AG) gas chromatograph equipped with a thermal conductivity detector, with high-purity argon as the carrier gas.

## 3. Result and Discussion

TiO_2_/g-C_3_N_4_ heterostructures were obtained by the electrostatic assembly of TiO_2_ nanoparticles and ultrathin g-C_3_N_4_ nanosheets (Figure 1, see Experimental section for details). Colloidal TiO_2_ nanoparticles were produced in the presence of OAm and OAc using TiF_4_ as the Ti precursor. As shown in Figure 2a, low-resolution TEM images exhibited the TiO_2_ particles to have a flat square morphology with a side length of 30–50 nm and a thickness of about 5–10 nm. g-C_3_N_4_ nanosheets were produced by the thermal etching of bulk g-C_3_N_4_. As observed by scanning electron microscopy (SEM, Appendix A) and transmission electron microscopy (TEM, Figure 2b) characterization, bCN and uCN displayed significantly different morphologies. The uCN showed a thin nanosheet-based structure pointing at the occurrence of a layer etching during the thermal process. Appendix A displays the nitrogen adsorption–desorption isotherms of bCN and uCN, which further proved uCN (85.7 m^2^/g) to be characterized by a larger specific surface area than bCN (46.3 m^2^/g).

To positively charge the surface of the TiO_2_ particles, enable their dispersion in an aqueous solution and promote charge transfer with the media; the organic ligands attached to the particle surface were removed using HBF_4_ (Appendix A). As observed by zeta-potential analysis, while the g-C_3_N_4_ nanosheets were negatively charged (V = −33.8 mV), after ligands removal the TiO_2_ particles were positively charged (V = +18.6 mV), which enabled the electrostatic self-assembly of the two components [38]. Indeed, when combining solutions of the two types of material, a light-yellow precipitate was formed. The precipitate was composed of large uCN nanosheets containing numerous nanoparticles attached to their surface. TEM analyses showed these nanoparticles lie flat on the surface of uCN, forming 2D/2D heterostructures (Figure 2c,d). High resolution TEM (HRTEM) further confirmed these nanoparticles are TiO_2_ with good crystallinity (Figure 2e,f).

SEM-EDS elemental maps (Appendix A) displayed a homogeneous distribution of C, N, O and Ti, demonstrating a uniform distribution of TiO_2_ particles on the uCN surface at the microscale. On the other hand, quantitative EDX analyses showed the TiO_2_:CN weight ratio to be close to that of the nominal combination of each phase: TiO_2_:CN = 0.47 for T_1_/uCN_2_; TiO_2_:CN = 1.1 for T_1_/uCN_1_ and TiO_2_:CN = 1.9 for T_2_/uCN_1_, obtained from mixing 1:2, 1:1 and 2:1 mass ratios of particles, respectively (Appendix A).

Figure 3a displays the X-ray diffraction (XRD) patterns of bCN, uCN, TiO_2_ and T/uCN samples. The XRD peaks at 25.2° (101), 38.0° (004), 47.7° (200) and 54.8° (211) are associated with the anatase TiO_2_ phase (JCPDS No. 21-1272) [39]. Additonally, the characteristic diffraction peaks at 13.1° and 27.4° correspond to the (002) and (100) planes of g-C_3_N_4_ (JCPDS No. 87-1526) [40]. The characteristic diffraction peaks of both TiO_2_ and g-C_3_N_4_ can be observed in all the composites samples, confirming the coexistence of anatase TiO_2_ and g-C_3_N_4_.

The X-ray photoelectron spectroscopy spectra of TiO_2_, uCN and T/uCN are displayed in Figure 3b–f. As observed from the survey XPS spectrum, besides Ti, C, O and N, a residual amount of F from the TiF_4_ precursor used to prepare the TiO_2_ particles was also present in the final material (Figure 3b). The high-resolution C 1s XPS spectrum of uCN showed two main contributions at 288.2 eV and 284.8 eV, which were assigned to C-(N3) and C–C/C=C, respectively (Figure 3c). Compared with pure uCN, the peak for C-(N3) of the T_1_/uCN_1_ sample was slightly shifted to 288.2 eV. The high-resolution N 1s XPS spectra were deconvoluted using three contributions at binding energies of 398.1 eV, 499.4 eV and 400.5 eV for uCN and 398.1 eV, 499.6 eV and 400.7 eV for T/uCN (Figure 3d). These three contributions were assigned to N-(C_2_), N-(C_3_) and N-H_x_ groups of the heptazine framework. The small shifts detected for C and some of the N components might be related to a certain degree of charge between the TiO_2_ and the CN phases. Figure 3e displays the high-resolution Ti 2p XPS spectra of TiO_2_ and T/uCN. Both samples show two strong peaks at approximately 458.7 and 464.5 eV, which are assigned to the Ti 2p_3/2_ and Ti 2p_1/2_ levels of Ti within a TiO_2_ environment. The high-resolution O 1s XPS spectra of TiO_2_ and T/uCN were fitted with two peaks at 530.4 eV and 531.8 eV, which were associated with oxygen within the TiO_2_ lattice and oxygen-containing surface adsorption groups such as surface hydroxyl, respectively (Figure 3f).

The UV-vis spectra showed the UV absorption edge of TiO_2_ particles and uCN nanosheets at about 390 nm and 445 nm, respectively (Figure 4a). T/uCN composites showed a similar onset absorption edge as uCN but an increased absorption below 400 nm related to the presence of the TiO_2_ component. All TiO_2_ and T/uCN samples presented a small absorption in the range 500–800 nm related to a small amount of F ion doping. According to the Kubelk–Munk function, the band gaps of TiO_2__,_ uCN and T_1_/Ucn_1_ samples were calculated at about 3.02 eV, 2.62 eV and 2.65 eV, respectively (Figure 4b).

According to Mott–Schottky analyses (Figure 4c,d and Appendix A), the flat band potentials of TiO_2_ and uCN were −0.36 V and −0.86 V vs. the normal hydrogen electrode (NHE). The valence band (VB) XPS spectra of TiO_2_ and uCN showed the valence band maximum (VBM) to be located at 2.89 eV and 2.46 eV from the Fermi level, respectively. Since the flat band potentials are approximately equal to the Fermi level [41,42], the VBM was located at 2.53 eV and 1.60 eV with respect to the NHE for TiO_2_ and uCN, respectively. Then, taking into account the calculated band gaps (E_g_ = E_vb_ − E_cb_) [43], the conduction band minimum (CBM) was located at 0.49 and −1.02 for TiO_2_ and uCN, respectively. Figure 4f displays the energy-level diagram calculated for TiO_2_ and uCN samples. According to this scheme, when combining uCN with TiO_2_, a type II heterojunction is formed, involving electron transfer from the uCN to the TiO_2_ particles. Besides, it is predicted that within such heterostructure, photogenerated electrons move toward the TiO_2_ phase and photogenerated holes toward the uCN, respectively.

To analyze the photocatalytic activity towards hydrogen generation, all the samples were loaded with 1 wt% platinum as cocatalyst. Figure 5 displays the photocatalytic hydrogen generation from bCN, uCN, TiO_2_ and TiO_2_/uCN composites for 4 h under simulated solar light and using methanol as a sacrificial agent. Appendix A show the chromatogram plots and the linear fitting of the standard hydrogen curve for gas chromatography, which show our measurement error is less than 0.2%.

For TiO_2_, a high hydrogen evolution rate (HER) up to 1449 μmol·g^−1^·h^−1^ was obtained. Additionally, a notable HER was also obtained from uCN (801 μmol·g^−1^·h^−1^), well above that of bCN (599 μmol·g^−1^·h^−1^), which is consistent with the larger surface area provided by the thin-layered structure of uCN. All the TiO_2_/uCN composites displayed a significant HER improvement with respect to pure TiO_2_ or uCN. The highest HERs were obtained with the TiO_2_/uCN composites having a 1:1 weight ratio of the two components, reaching a HER of 3875 μmol·g^−1^·h^−1^, which is 2.7 and 4.8 times higher than that of TiO_2_ and uCN, respectively. The observed synergistic effect obtained when mixing both materials is related to the transfer and thus separation of photogenerated carriers at the 2D/2D heterojunctions, which prevents their recombination. Appendix A provides a comparison of the activity obtained here with those of previous published works, demonstrating the outstanding activity provided by the 2D/2D TiO_2_/uCN heterojunction.

As a reference, we also measured the HER of TiO_2_/bCN composites with the optimized weight ratio 1:1 (T_1_/bCN_1_). As observed in Figure 5c and Appendix A, the HER of T_1_/bCN_1_ also showed an obvious improvement with respect to that of pure TiO_2_ and bCN, but the highest HER values were well below those of 2D/2D T/uCN heterojunctions having extended surface and interface areas.

The apparent quantum yield (AQY) of the process was evaluated under 380 nm (4.51 mW·cm^−2^) and 420 nm (12.14 mW·cm^−2^) irradiation (Appendix A, see details in Appendix A). For T_1_/uCN_1_, the AQY at 380 nm and 420 nm was estimated at 7.61% and 2.64%, respectively, which is consistent with UV-vis spectroscopy results (Figure 5d).

Figure 6a displays the positive photocurrents measured from uCN, TiO_2_ and TiO_2_/uCN samples under simulated solar irradiation. All the composite T/uCN, electrodes displayed significantly higher photocurrents than pure TiO_2_ and uCN, especially the T_1_/uCN_1_ electrode that showed the highest photocurrents, fourfold higher than those of uCN and TiO_2_. This result further confirms an improvement of the charge separation/transport with the formation of the 2D/2D heterojunction.

Electrochemical impedance spectroscopy (EIS) was further employed to identify the charge transport dynamics. Figure 6b displays the Nyquist plot of the impedance spectra of TiO_2_, uCN and T_1_/uCN_1_. Consistent with previous results, the T_1_/uCN_1_ electrode presented a much smaller arc radius than the other two samples, confirming a much lower charge transfer resistance with the formation of the 2D/2D TiO_2_/uCN heterojunction [44].

A strong photoluminescence (PL) peak was obtained under 370 nm light excitation from the uCN sample at about 455 nm, which is ascribed to the radiative band-to-band recombination of photogenerated charge carriers. When incorporating increasing amounts of TiO_2_, the PL intensity of T/uCN was progressively quenched (Appendix A). Additional time-resolved PL (TRPL) spectra under 365 nm light excitation (Figure 6c) allowed calculating significantly longer PL lifetimes (4.72 ns) for T_1_/uCN_1_ samples than for TiO_2_ (3.15 ns) and uCN (3.51 ns), which points at an effective separation of photogenerated charge carriers within the TiO_2_/uCN heterostructures [45].

Based on the above results, the photocatalytic mechanism displayed in Figure 7 is proposed for hydrogen generation in T/uCN heterojunction photocatalysts. While both TiO_2_ and uCN can generate electrons and holes under simulated solar light irradiation, the photogenerated electron–hole pairs in pure TiO_2_ and uCN rapidly recombine, resulting in moderate HERs. Through the formation of a 2D/2D T/uCN heterostructure, the photogenerated electrons remain or are transferred to the TiO_2_ CB because the TiO_2_ CBM is located 0.53 eV below that of CN. Similarly, photogenerated holes remain or are driven to the uCN VB, which is located 0.93 eV above that of TiO_2_. Electrons at the TiO_2_ CB migrate to the platinum, which has a larger work function, thus a lower Fermi level, from where they are transferred to adsorbed H^+^ to produce H_2_. On the other hand, holes react with sacrificial methanol at the CN surface. Consequently, the photocatalytic hydrogen evolution process using sacrificial methanol can be described as follows:(1)$$\mathrm{uCN}/{\mathrm{TiO}}_{2}\overset{hv}{\to }\mathrm{uCN}/{\mathrm{TiO}}_{2}\left(e^{-}+h^{+}\right)$$
(2)$$\mathrm{uCN}/{\mathrm{TiO}}_{2}\left(e^{-}+h^{+}\right)\to \mathrm{uCN}\left(h^{+}\right)+{\mathrm{TiO}}_{2}\left(e^{-}\right)$$
(3)$${\mathrm{TiO}}_{2}\left(e^{-}\right)+\mathrm{Pt}\to {\mathrm{TiO}}_{2}+\mathrm{Pt}\left(e^{-}\right)$$
(4)$$2H^{+}+e^{-}\to H_{2}$$
(5)$$2h^{+}+{\mathrm{CH}}_{3}\mathrm{OH}\to {\mathrm{CH}}_{2}O+2H^{+}$$

Finally, the stability of the T_1_/uCN_1_ photocatalyst in hydrogen evolution conditions under simulated solar light irradiation was measured through five four-hour cycles. As shown in Appendix A, after this 20 h of reaction, the photocatalytic performance was hardly reduced, proving the excellent stability and reusability of the T_1_/uCN_1_ photocatalyst. Additionally, as displayed in Appendix A, SEM and XRD analysis of the catalyst after 20 h photocatalytic hydrogen generation reaction demonstrated the morphology and crystallographic structure of the material to be stable under photocatalytic reaction conditions.

## 4. Conclusions

In summary, we detailed the synthesis of 2D/2D T/uCN heterojunctions from ultrathin g-C_3_N_4_ (uCN) and colloidal TiO_2_ nanosheets through an electrostatic self-assembly approach. The highest hydrogen generation rate was achieved from T/uCN composites with a 1:1 mass ratio of the two components. The photocatalytic performance for H_2_ production was increased in the following order: bCN < uCN < TiO_2_ < T_1_/uCN_2_ < T_2_/uCN_1_ < T_1_/uCN_1_. The enhanced performance was attributed to the unique 2D/2D type II heterojunction architecture that simultaneously maximized the surface area to interact with the media and the interface between the two materials. The face-to-face interfacial contact between ultrathin layers of g-C_3_N_4_ and the faceted TiO_2_ provided fast separation of photogenerated charges inside the composites, reducing recombination and thus increasing the apparent quantum yield.

## Figures and Tables

**Figure 1 nanomaterials-12-01557-f001:**
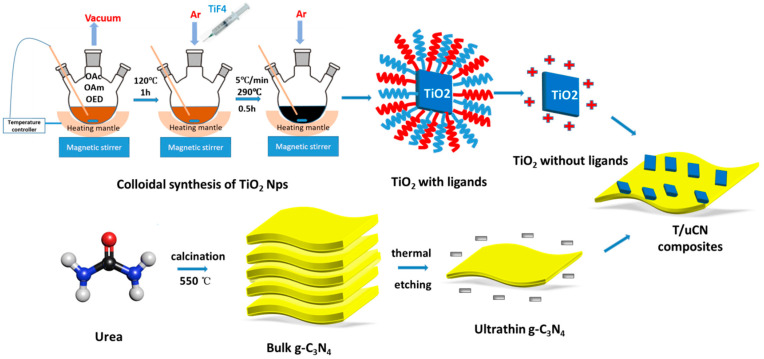
Schematic illustration of the process used to produce 2D/2D TiO_2_/uCN composite.

**Figure 2 nanomaterials-12-01557-f002:**
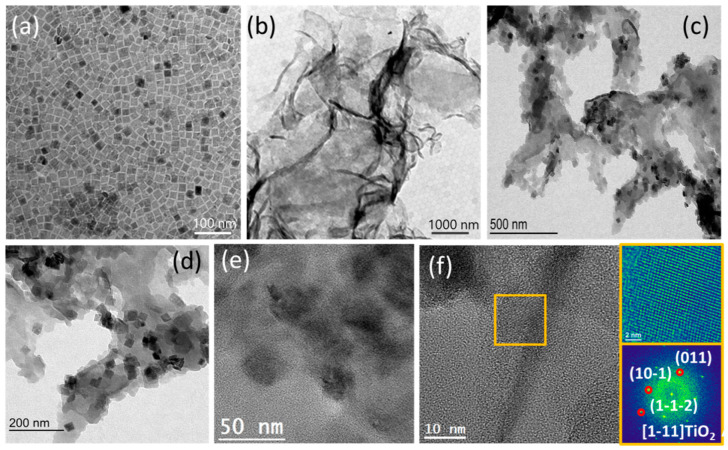
Representative TEM images of (**a**) TiO_2_ nanoparticles; (**b**) g-C_3_N_4_ nanosheets and T_1_/uCN_1_ composite with representative (**c**) low and (**d**) high magnification. (**e**,**f**) HRTEM images of T_1_/uCN_1_. A magnified detail (**top right**) of the orange squared region in the HRTEM image and its corresponding indexed power spectrum (**bottom right**) is shown, revealing the TiO_2_ anatase phase (space group = I4_1_/amd) with a = b = 3.7840 Å, and c = 9.5000 Å. TiO_2_ lattice fringe distances were measured to be 0.233 nm, 0.352 nm and 0.348 nm at 41.30° and 139.38°, which could be interpreted as the anatase TiO_2_ phase, visualized along its [1–11] zone axis.

**Figure 3 nanomaterials-12-01557-f003:**
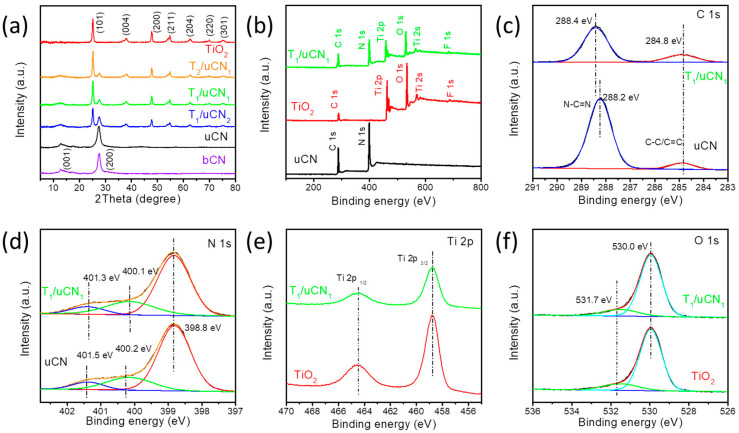
(**a**) XRD patterns of TiO_2_, uCN and T/uCN. (**b**) XPS survey spectrum of TiO_2_, uCN and T/uCN; high-resolution XPS spectra at the regions (**c**) C 1s, (**d**) N 1s, (**e**) Ti 2p and (**f**) O 1s.

**Figure 4 nanomaterials-12-01557-f004:**
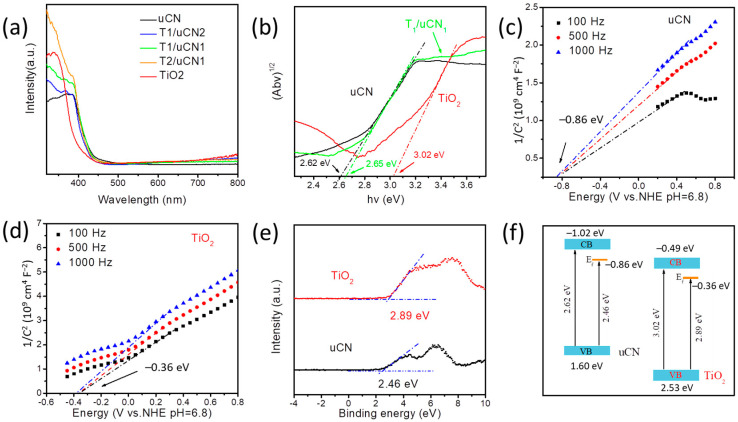
(**a**) UV-vis absorption spectra. (**b**) Kubelka–Munk-transformed function of TiO_2_, uCN and T_1_/uCN_1_. (**c**,**d**) Mott–Schottky plots of uCN (**c**) and TiO_2_ (**d**). (**e**) Valence band XPS spectrum of TiO_2_ and uCN. (**f**) Diagram of the band structure of TiO_2_ and uCN.

**Figure 5 nanomaterials-12-01557-f005:**
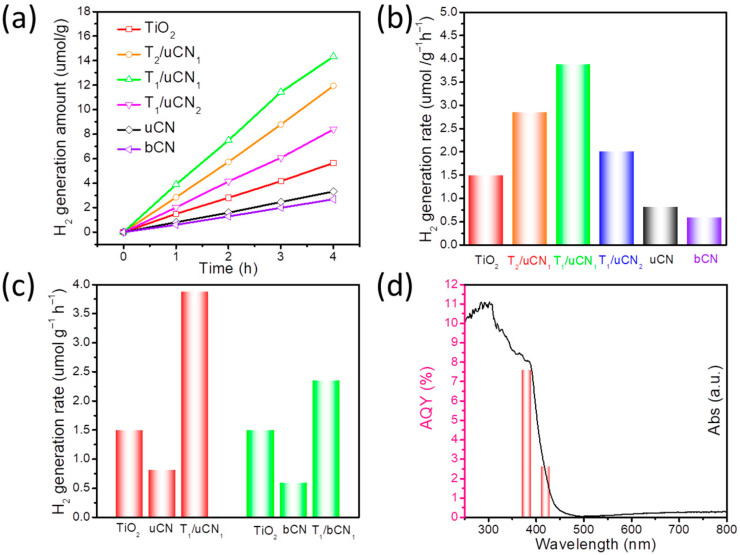
(**a**) Photocatalytic hydrogen generation on bCN, TiO_2_ and T/uCN samples during four hours under simulated solar light illumination. (**b**) Photocatalytic hydrogen peroxide generation rate of bCN, TiO_2_ and T/uCN samples. (**c**) H_2_ production rate contrast between T_1_/uCN_1_ and T_1_/bCN_1_. (**d**) Wavelength-dependent AQY of T_1_/uCN_1_.

**Figure 6 nanomaterials-12-01557-f006:**
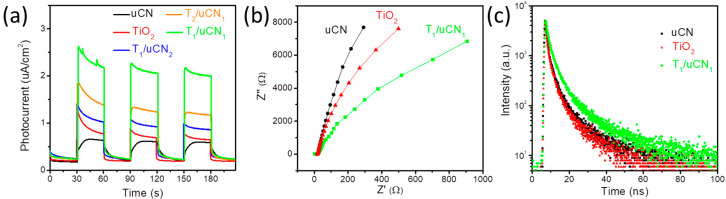
(**a**) Photocurrent response curves of bCN, TiO_2_ and T/uCN samples; (**b**) electrochemical impedance spectroscopy (EIS) Nyquist plots of bCN,TiO_2_ and T_1_/uCN_1_ sample; (**c**) TRPL decay of bCN, TiO_2_ and T/uCN samples.

**Figure 7 nanomaterials-12-01557-f007:**
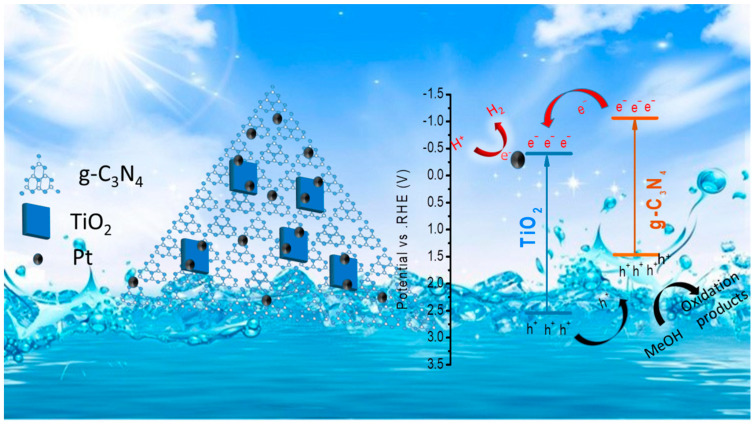
Schematic diagram of photocatalytic hydrogen production over T/uCN photocatalyst.

## Data Availability

The data are available on reasonable request from the corresponding authors.

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
