# Peer review of "2D/2D Heterojunction of TiO2 Nanoparticles and Ultrathin G-C3N4 Nanosheets for Efficient Photocatalytic Hydrogen Evolution"

_nanomaterials, 2022, doi:10.3390/nano12091557_

Round 1
Reviewer 1 Report
The present work is a systematic study of composite structured photocatalysts based on titanium dioxide and graphite-like carbon nitride. The work is well structured, the samples are characterized by a complex of physicochemical methods, and the setting up of kinteic experiments is beyond doubt. It is very important that the high stability of the synthesized samples is shown. While reading, I had only a number of minor remarks.
- Experimental part. The authors write about the deposition of platinum before the hydrogen production process, but mention HAuCl4. What ended up being deposited, platinum or gold?
- Figure 5. The error of measurements should be shown.
- Figure 7. The authors make the assumption that heterojunctions are implemented according to Scheme II, while now most scientists are inclined to believe that heterojunctions according to Z- and S-schemes are more efficient.
Author Response
T

Reviewer 2 Report
The work reported in the manuscript deals with the fabrication of 2D/2D GCN/TiO2 heterostructures for photocatalytic H2 evolution reaction. In the current work, authors first prepared GCN and then converted to uCN via secondary calcination approach. Later authors mixed with TiO2 nanosheets which were separately prepared and loaded on to uCN via electrostatic self-assembly method. Finally the obtained GCN/TiO2 heterostructures were utilized as visible photocatalysts for H2 evolution reactions. Results were analyzed and presented carefully. Overall, the quality of manuscript is good and suitable for Nanomaterials journal. Before publication, authors need to address the following issues,
- Introduction part is poor. Authors should modify the introduction to highlight the objective of current work. This section can be further improved by citing latest visible active photocatalyst works such as: (a) Nanomaterials 2020, 10(4), 619; (b) Applied Catalysis B: Environmental Volume 248, 5 July 2019, Pages 538-551; (c) Applied Surface Science Volume 565, 1 November 2021, 150601
- Are the authors performed any specific surface area measurements. It is advised to provide if possible
- Check Fig. 2 captions, (a-d) is for TEM images of TiO2 ?
- It is advised to compare the activity of photocatalyst in the current work with the reported results.
Author Response
The work reported in the manuscript deals with the fabrication of 2D/2D GCN/TiO2 heterostructures for photocatalytic H2 evolution reaction. In the current work, authors first prepared GCN and then converted to uCN via secondary calcination approach. Later authors mixed with TiO2 nanosheets which were separately prepared and loaded on to uCN via electrostatic self-assembly method. Finally the obtained GCN/TiO2 heterostructures were utilized as visible photocatalysts for H2 evolution reactions. Results were analyzed and presented carefully. Overall, the quality of manuscript is good and suitable for Nanomaterials journal. Before publication, authors need to address the following issues,
- Introduction part is poor. Authors should modify the introduction to highlight the objective of current work. This section can be further improved by citing latest visible active photocatalyst works such as: (a) Nanomaterials 2020, 10(4), 619; (b) Applied Catalysis B: Environmental Volume 248, 5 July 2019, Pages 538-551; (c) Applied Surface Science Volume 565, 1 November 2021, 150601
- Are the authors performed any specific surface area measurements. It is advised to provide if possible
- Check Fig. 2 captions, (a-d) is for TEM images of TiO2 ?
- It is advised to compare the activity of photocatalyst in the current work with the reported results.
We thank the reviewer for carefully reading the manuscript and for providing valuable comments and suggestions that certainly helped to improve the work.
- Following the reviewer suggestion, we clarify the work objective and introduce the latest photocatalytic work as followï¼›
In the present work, we target improving photocatalytic hydrogen production using 2D/2D heterojunctions.
Coupling with other semiconductors to form heterostructures. [16–24]
Graphite carbonitride (g-C3N4) with a layered structure similar to graphite, high chemical stability and low cost has received increasing interest in recent years.[25–27]
[24] R.K. Chava, N. Son, M. Kang, Surface engineering of CdS with ternary Bi/Bi2MoO6-MoS2 heterojunctions for enhanced photoexcited charge separation in solar-driven hydrogen evolution reaction, Appl. Surf. Sci. 565 (2021) 150601.
[27] R.K. Chava, J. Do, M. Kang, Strategy for improving the visible photocatalytic H2 evolution activity of 2D graphitic carbon nitride nanosheets through the modification with metal and metal oxide nanocomponents, Appl. Catal. B Environ. 248 (2019) 538–551.
- As suggested by the reviewer, We add the specific surface area data to SI,and gave some discussion in manuscript at page as follow:
Figure S1c displays the nitrogen adsorption-desorption isotherms of bCN and uCN, which further proved uCN (85.7m2/g) has a large specific surface area than bCN (46.3m2/g).
Fig.S1 SEM image of (a) bulk g-C3N4 and (b) ultrathin g-C3N4, (c) N2 adsorption-desorption isotherms of bCN and uCN.
- Thank the reviewer comment,we modified the captions of Fig. 2 as follow;
Figure 2. Representative TEM images of (a) TiO2 nanoparticles; (b) g-C3N4 nanosheets; and T1/uCN1 composite with representative (c) low and (d) high magnification.
- Following the reviewer suggestion, we added a table of photocatalytic hydrogen production about TiO2/g-C3N4 based catalysts on Table S3.
Table S3. photocatalytic hydrogen production about TiO2/g-C3N4 based catalysts

Reviewer 3 Report
Authors of this work presents a electro-static self-assembly method for production of ultrathin 2D g-C3N4 nanosheets coupled with flat TiO2 nanoparticles as face-to-face 2D/2D heterojunction. Investigated structures exhibit effective charge separation and transport properties that translate into out-standing photocatalytic performance. The experimental part is well-written, some of experimental details presented in supplementary information file. The level this work is high.
My several remarks a listed below:
Page 4, Figure 2.
The Figure 2a is not mentioned in the text. It seems like it should be mentioned in this sentence: Low-resolution TEM images exhibited the TiO2 particles to have a flat square morphology with a side length of 30-50 nm and a thickness of about 5-10 nm.
Supplementary file, Figure S2
Infrared spectra of TiO2 nanoparticles after removing ligands is totally flat. This is good, but where is the peaks related to Ti-O vibrations?
Page 6, Figure 4b
But where are the Kubelka-Munk function curves for heterostructures?
Author Response
The present work is a
Authors of this work presents a electro-static self-assembly method for production of ultrathin 2D g-C3N4 nanosheets coupled with flat TiO2 nanoparticles as face-to-face 2D/2D heterojunction. Investigated structures exhibit effective charge separation and transport properties that translate into out-standing photocatalytic performance. The experimental part is well-written, some of experimental details presented in supplementary information file. The level this work is high.
My several remarks a listed below:
Page 4, Figure 2.
The Figure 2a is not mentioned in the text. It seems like it should be mentioned in this sentence: Low-resolution TEM images exhibited the TiO2 particles to have a flat square morphology with a side length of 30-50 nm and a thickness of about 5-10 nm.
Following the reviewer suggestion, the above sentence were modified as follow:
As shown in Figure 2a,low-resolution TEM images exhibited the TiO2 particles to have a flat square morphology with a side length of 30-50 nm and a thickness of about 5-10 nm.
Supplementary file, Figure S2
Infrared spectra of TiO2 nanoparticles after removing ligands is totally flat. This is good, but where is the peaks related to Ti-O vibrations?
Thanks for reviewer comments. There should be an IR peak of Ti-O around 483cm-1 and also some oxygen-containing functional groups at positions over 3000 cm-1, but these peak was overlooked probably due to the low concentration of nanoparticles in the solution we used to measure. We re-tested with solid samples and brought more accurate results as follow:
Figure S2. FTIR spectra of OAC, OLMA and TiO2 before and after ligands remove.
Page 6, Figure 4b
But where are the Kubelka-Munk function curves for heterostructures?
The Kubelka-Munk function curves are very similar to that of uCN. Still, according to the reviewer suggestion, we included the curve obtained from T1/uCN1 sample in fig 4b.
Figure 4. (a) UV-vis absorption spectra. (b) Kubelka-Munk transformed function of TiO2, uCN and T1/uCN1. (c,d) Mott–Schottky plots of uCN (c) and TiO2 (d). (e) Valence band XPS spectrum of TiO2 and uCN. (f) Diagram of the band structure of TiO2 and uCN.

Round 2
Reviewer 2 Report
The revisions made to the manuscript are acceptable and hence the manuscript is suitable for publication.